# Comparison of Frailty Assessment Tools for Older Thai Individuals at the Out-Patient Clinic of the Family Medicine Department

**DOI:** 10.3390/ijerph20054020

**Published:** 2023-02-23

**Authors:** Pimonpan Rattanapattanakul, Adchara Prommaban, Peerasak Lerttrakarnnon

**Affiliations:** 1Geriatric Medical Center, Excellent Center, Faculty of Medicine, Chiang Mai University, Chiang Mai 50200, Thailand; 2Aging and Aging Palliative Care Research Cluster, Department of Family Medicine, Faculty of Medicine, Chiang Mai University, Chiang Mai 50200, Thailand

**Keywords:** frailty, frailty assessment tools, out-patient, older patients, Thai older patients

## Abstract

This study evaluated the validity of the screening tools used to evaluate the frailty status of older Thai people. A cross-sectional study of 251 patients aged 60 years or more in an out-patient department was conducted using the Frailty Assessment Tool of the Thai Ministry of Public Health (FATMPH) and the Frail Non-Disabled (FiND) questionnaire, and the results were compared with Fried’s Frailty Phenotype (FFP). The validity of the data acquired using each method was evaluated by examining their sensitivity, specificity, positive predictive value (PPV), negative predictive value (NPV), and Cohen’s kappa coefficient. Most of the participants were female (60.96%), and most were between 60 and 69 years old (65.34%). The measured prevalences of frailty were 8.37%, 17.53%, and 3.98% using FFP, FATMPH, and FiND tools, respectively. FATMP had a sensitivity of 57.14%, a specificity of 86.09%, a PPV of 27.27%, and an NPV of 95.65%. FiND had a sensitivity of 19.05%, a specificity of 97.39%, a PPV of 40.00%, and an NPV of 92.94%. The results of the Cohen’s kappa comparison of these two tools and FFP were 0.298 for FATMPH and 0.147 for FiND. The predictive values of both FATMPH and FiND were insufficient for assessing frailty in a clinical setting. Additional research on other frailty tools is necessary to improve the accuracy of frailty screening in the older population of Thailand.

## 1. Introduction

Frailty is defined as a reduction in the ability to cope with everyday or acute stressors, particularly among older adults [1]. Frailty results in an increased vulnerability brought about by age-associated declines in physiological reserves and functioning across multiple organ systems [1]. The consequences of this condition heighten an individual’s susceptibility to increased dependency and vulnerability, as well as to an increased risk of death [1,2]. The health care system is affected by increases in health care needs, admissions to hospital, and admissions to long-term care. However, frailty is a dynamic process which can emerge from pre-frail or robust statuses [3]. Validated assessment tools and appropriate interventions are important to reduce morbidity and mortality. A systematic review and meta-analysis of a survey of the models used to evaluate frailty among ≥ 50-year-olds in 62 countries found that 12% of prevalence used physical frailty models and 24% used deficit accumulation models. The prevalences of the consideration of pre-frailty were 46% and 49% for the physical frailty models and the deficit accumulation models, respectively [4]. In terms of geographical location, using physical frailty models, the highest prevalence of physical frailty was found in Africa (22%) and the lowest prevalence was in Europe (8%), while the pre-frailty prevalence was highest in the Americas (50%) and lowest in Europe (42%). However, using deficit accumulation models, the prevalence of frailty was found to be highest in Oceania (31%) and lowest in Europe (22%), while pre-frailty prevalence was highest in Oceania (51%) and lowest in Europe and Asia (49%). The population-level frailty prevalence among community-dwelling adults varied by age, gender, and frailty classification [4].

Several studies have reported that frailty is related to a variety of negative health outcomes and diseases. In 2013, cognitive frailty was described as a group of heterogeneous clinical symptoms based on the presence of both physical frailty and cognitive impairment, excluding consistent Alzheimer’s disease or other dementias. The prevalence of cognitive frailty among community-dwelling older adults was reported to be 9% in a systematic review and meta-analysis [5]. Similarly, the prevalences of frailty and pre-frailty were found to be 20.1% and 49.1%, respectively, in a systematic review and meta-analysis study of community-dwelling older adults with diabetes. Older adults with diabetes were more susceptible to being frail than those without diabetes [6]. Additional factors were found to have an influence on frailty; for example, fruit and vegetable consumption was associated with a lower risk of frailty [7].

There are many measurement tools available which can provide frailty scores when used to screen for or assess the degree of frailty; however, no single score metric is considered the gold standard [2,8]. It has been recommended that geriatricians in the Asia-Pacific region use a validated measurement tool to identify frailty [2]. There are three major approaches used, i.e., the physical frailty phenotype model of Fried et al. and its rapid screening tool, FRAIL; the deficit accumulation model of Rockwood and Mitnitski, which captures multimorbidity; and mixed physical and psychosocial models, such as the Tilburg Frailty Indicator [9] and the Edmonton Frailty Scale [10]. Another approach by Aguayo GA et al. [8] consists of the use of four models, including a phenotype of the frailty model, a multidimensional model, an accumulation of deficits model, and a disability model. 

The most commonly used method in the literature is the physical frailty phenotype [11]. The phenotype diagnosis is based on three of the following five criteria: weight loss, exhaustion, physical inactivity, slow walking speed, and weak grip strength [12]. The present study reviews five phenotypic criteria that have been measured in different ways across various studies which could potentially affect the estimates of the prevalence of frailty and the predictive ability of the aforementioned phenotype, potentially leading to different classifications and results [11]. Kutner and Zhang [13] commented on the replacement of the performance-based measures (i.e., grip strength and walking speed) in the original frailty phenotype definition with self-reported items. 

In Thailand, a study by Boribun N. et. al. [14] found that the prevalence of frailty in Thai community-dwelling older adults was 24.6%, based on the Frail Non-Disabled (FiND) questionnaire. A 2020 study by Sukkriang and Punsawad [15], which used various frailty assessment tools, found that the prevalence of frailty of older individuals in Thai communities was 11.7%, using Fried’s Frailty Phenotype (Cardiovascular Heart Study) criteria, and studied the validity of various frailty assessment tools. The Clinical Frailty Scale (CFS) used in the same study had a sensitivity of 56% and a specificity of 98.41%; the simple FRAIL questionnaire had a sensitivity of 88% and a specificity of 85.71%; the PRISMA-7 questionnaire sensitivity was 76%; and the specificity was 86.24%. The Timed Up and Go (TUG) test had a sensitivity of 72% and a specificity of 82.54%. The Gerontopole frailty screening tool (GFST) sensitivity was 88% and the specificity was 83.56%. The study by Sriwong et al. (2022) [16] developed a Thai version of the Simple Frailty Questionnaire (T-FRAIL) and modified it to improve its diagnostic properties in the preoperative setting. Their study found that the incidence of frailty diagnosed using the Thai Frailty Index was 40.0%. The identification of frailty using a score of two points or more provided the best Youden index, at 63.1, with a sensitivity of 77.5% (95% CI 69.0–84.6) and a specificity of 85.6% (95% CI 79.6–90.3). 

There is currently a need for simple, valid, accurate, and reliable methods and tools for detecting frailty which are appropriate for the Thai population. Our team works in an academic hospital and has developed evidence data in our clinic in the hospital. Therefore, the present study was conducted in this clinic. This study compared selected frailty assessment tools, including Fried’s Frailty phenotype (FFP), which is the most commonly used assessment tool used for reference; the Frailty Assessment Tool of the Thai Ministry of Public Health (FATMPH), which is recommended in the Thai check-up manual but lacks published validation; and the FiND questionnaire, which is used in communities but, as yet, there is no evidence of its use at the Out-Patient Department (OPD) of Maharaj Nakorn Chiang Mai Hospital (a university-level hospital).

## 2. Materials and Methods

### 2.1. Samples

This cross-sectional study included 251 older patients (age 60 years or older) who came to the OPD of the Family Medicine Department, Maharaj Nakorn Chiang Mai Hospital, Faculty of Medicine, Chiang Mai University, during the period of December 2016–March 2017. The patients signed a consent form declaring their agreement to participate in this research. This study was approved by the Research Ethics Committee of the Faculty of Medicine of Chiang Mai University (no. 380/2016). The inclusion criteria for participants were: (1) Thais 60 years or older and who had been seen at the OPD for more than 1 year, (2) the ability to communicate orally in Thai and read the Thai language, (3) the ability to walk by themselves or with walking aids. The exclusion criteria were: (1) being bed ridden, (2) being handicapped in both hands, (3) currently having a serious illness, and (4) having impaired cognition. 

The sample size was calculated to be 230 using the following formula:*n* = Z2α/2 × Se(1 − Se)/d2 × Prev
where *n* = sample size, Se = sensitivity (0.9), Prev = prevalence (0.15) [17], d = precision of the estimate (1.0), and alpha = 0.1.

### 2.2. Frailty Assessment Tools 

#### 2.2.1. Fried’s Frailty Phenotype

The five criteria of Fried’s Frailty Phenotype (FFP) assessment were used as the reference assessment tool in this study, following Fried et al. [12], with slight modification. These criteria were:(1)Weight loss. My weight has decreased at least 4.5 kg in the past year or I have had an unintentional weight loss of at least 5% of my previous year’s body weight (no = 0, yes = 1).(2)Exhaustion. Self-reported results of the Center for Epidemiologic Studies Depression scale (CES–D). Two statements were provided: (2.1) I felt that everything I did was an effort and (2.2) I could not get going. The question is then asked, “How often in the last week did you feel this way?” The alternative answers are: 0 = rarely or none of the time (<1 day), 1 = some or a little of the time (1–2 days), 2 = a moderate amount of the time (3–4 days), or 3 = most of the time. Answers of “2” or “3” to either of these questions were categorized as frail by the exhaustion criterion (no = 0, yes = 1).(3)Slowness. My walking speed is 20% below baseline (adjusted for gender and height) (no = 0, yes = 1).(4)Weakness. Grip strength is 20% below baseline (adjust for gender and body mass index) (no = 0, yes = 1).(5)Low activity was evaluated with the following question: How often do you engage in activities that require a low or moderate amount of energy such as gardening, cleaning the car, or walking? (more than once a week = 1, once a week = 2, one to three times a month = 3 and hardly ever or never = 4) [18].

A combined FFP score of 0 was considered a “non-frail” phenotype; a score of 1 or 2 was considered a “pre-frail” phenotype; and a score of 3 or more was considered a “frail” phenotype.

#### 2.2.2. Frailty Assessment Tool of the Thai Ministry of Public Health

The Frailty Assessment Tool of the Thai Ministry of Public Health (FATMPH) is a modification of Fried’s Frailty Phenotype, and is included in the Elderly Screening/Assessment Manual (2015) [19]. The assessment tool has 5 criteria: four questions are self-reports and one is based on measurement by medical staff:(1)In the past year, has your weight has decreased by more than 4.5 kg? (no = 0, yes = 1)(2)Do you feel tired all the time? (no = 0, yes = 1)(3)Are you unable to walk alone and need someone for support? (no = 0, yes = 1)(4)The participants walked in a straight line for a distance of 4.5 m. Time was measured from when they started walking (Time < 7 s = 0, time ≥ 7 s or could not walk = 1).(5)The participant had an obvious weakness in their hands, arms, and legs (no = 0, yes = 1).

A FATMPH score of 0 was considered a phenotype of “non-frail”; a score of 1 or 2 was considered a phenotype of “pre-frail”; and a score of 3 or more was consider a phenotype of “frail”.

#### 2.2.3. Frail Non-Disabled (FiND) Questionnaire

The Frail Non-Disabled (FiND) questionnaire is designed to differentiate between frailty and disability. FiND was used for community-dwelling older Thai adults by Boribun N et. al. [14]. The content validity index (CVI) was 0.8 and Cronbach’s alpha was 0.89 [13]. The FiND questionnaire consists of 5 questions: Do you have any difficulty walking 400 m? (no or some difficulty = 0, much difficulty or unable = 1)Do you have any difficulty climbing up a flight of stairs? (no or some difficulty = 0, much difficulty or unable = 1)During the last year, have you involuntarily lost more than 4.5 kg? (no = 0, yes = 1)How often in the last week did you feel that everything you did was an effort or that you could not get going? (2 times or less = 0, 3 or more times = 1)What is your level of physical activity? (at least 2–4 h per week = 0, mainly sedentary = 1)

A combined score of A + B + C + D + E = 0 was considered as “non-frail”; A + B = 0 and C + D + E ≥ 1 was considered as “frail”; and A + B ≥ 1 was considered as “disabled”.

### 2.3. Data Collection 

Data were collected using questionnaires and assessed using various tools. The general characteristics recorded included age, sex, religion, education, income, source of payment of medical expenses, history of family disease, present weight, weight one year ago, height, and body mass index. All participants were assessed using the Thai-language version of FATMPH, FFP, and FiND. The inter-rater reliability was 1.0 between researchers and assistants.

### 2.4. Statistical Analysis

The data were analyzed using Stata 12.0 and are presented as frequency, percentage, mean, and standard deviation (SD). The frailty assessment tools were analyzed for their sensitivity, specificity, positive predictive value (PPV), and negative predictive value (NPV); Cohen’s kappa was used to measure the reliability of these assessment tools.

### 2.5. Evaluation Consequence

All frail participants who were involved in any of the study of the assessment tools were advised to undergo comprehensive geriatric assessment. The appropriate interventions were then provided to these individuals. 

## 3. Results

The demographic characteristics of the 251 older participants from the OPD are shown in Table 1. Most were female and ranged in age from 60 to 69. The majority of the participants were married or living with a partner, had lower than a high school education, and were Buddhist. Half the participants were government officials. Most participants had an income of more than 10,000 baht per month. Their major source of income was from pensions, which provided an adequate income. 

The health status of the participants is shown in Table 2. Several medical conditions were identified among the participants. The most prevalent was hypertension, followed (in declining order of incidence) by dyslipidemia, diabetes mellitus, hyperuricemia, glaucoma or cataracts, chronic kidney disease, benign prostatic hypertrophy, coronary artery disease, cerebrovascular disease, and malignancy, followed by others. 

In this study, frailty status was evaluated using frailty assessment tools including FFP, FATMPH, and FiND. The frail and non-frail phenotypes were defined based on the combined results of all the assessment tools. The study found that the overall prevalence of frailty was 8.37% based on FFP, most of whom were female (90.47%). The frailty phenotype prevalence determined using FATMPH was 17.53% (female = 65.91%); using FiND, the frailty phenotype prevalence determined was 3.98% (female = 80.00%) (Table 3 and Table 4). 

The sensitivity, specificity, positive predictive value, and negative predictive value of the FATMPH and FiND tools were analyzed and compared with the standard FFP tool. As shown in Table 5, FATMHP had a sensitivity of 57.14%, a specificity of 86.09%, a positive predictive value (PPV) of 27.27%, and a negative predictive value (NPV) of 95.65%. FiND had a sensitivity of 19.05%, a specificity of 97.39%, a PPV of 40.00% and an NPV of 92.94%. The comparison of FATMPH and FiND with FFP found the Cohen kappa statistics to be 0.298 for FATMPH and 0.147 for FiND.

## 4. Discussion

Fried’s Frailty Phenotype (FFP) is a well-known and regularly utilized tool for identifying frailty in older individuals [20]. In Thailand, FATMPH was developed as a frailty assessment tool based on FFP. Even though the Fried criteria were not initially intended to be used as a self-reported questionnaire, researchers now usually employ modified questionnaires based on this frailty phenotype [21,22]. The Frail Non-Disabled (FiND) questionnaire, a self-administered frailty screening instrument designed to differentiate frailty from disability, was developed as a screening tool [23]. We focused on the comparison of both FATMPH and FiND with FFP, which is currently used to assess older patients at the OPD of the Family Medicine Department of the Maharaj Nakorn Chiang Mai Hospital Faculty of Medicine. Most of the participants had a chronic disease (92.43%), most frequently hypertension (65.75%). The prevalence of frailty in this study was 8.37% using FFP, which is lower than the prevalence of frailty among community-based elderly people (9.9%) [24]. Differences in frailty prevalence were due at least in part to differences in the assessment tools used, as well as the different geographical locations covered in this study. Frailty prevalence increased with age and was higher for females than males [3]. The relatively low prevalence of frailty in the study may be due to the fact that most of the participants were in the younger group of the elderly participants (60–69 years, 65.34%).

A screening test is defined as a medical test or procedure performed on members (subjects) of a defined asymptomatic population or population subgroup to assess the likelihood of their members having a particular disease or condition [25]. A screening test has only two possible outcomes: positive, suggesting that the subject has the disease or condition; or negative, suggesting that the subject does not have the disease or condition [26]. In prior research, a Korean version of the FRAIL scale (K-FRAIL) was found to be consistent with the multidimensional frailty index and to be a concise tool for screening for frailty in a clinical setting in Korea [24].

In Thailand, many frailty assessment tools have been established for use both for community-dwelling individuals [14,27,28] and in hospitals [16,29]. There have, however, been few studies in Thailand that have included a comparison and validation of the frailty assessment tools used for older Thai adults in order to evaluate their diagnostic efficacy. A previous comparative study of the Thai version of the Simple Frailty Questionnaire (T-FRAIL) and the Thai Frailty Index (TFI) found that T-FRAIL was valid and reliable for frailty detection in elderly patients at a surgery out-patient clinic [16]. Another study of community-dwelling elderly compared several screening tests, including CFS, the simple FRAIL questionnaire, the PRISMA-7 questionnaire, the TUG, and the GFST with Fried’s Frailty Phenotype method. That study found the simple FRAIL questionnaire and the GFST were the most appropriate tests for screening frailty due to their high sensitivity [15].

The present study is the first study to compare the use of FATMPH and FiND with FFP regarding patients in an OPD for older Thai adults. The comparison of FATMPH and FiND found that the sensitivity of FATMPH (57.14%) was higher than that of FiND (19.05%), but that the specificity of FATMPH (86.09%) was lower than that of FiND (97.39%). FATMPH and FiND were both had a lower sensitivity than CFS (56%), the simple FRAIL questionnaire (88%), the PRISMA-7 questionnaire (76%), the TUG (72%), and the GFST (88%), as reported in the study by Sukkriang and Punsawad [15], as well as the modified T-Frails, including T-Frail M1 (83.3%) and T-Frail M2 (85.8%), as reported in the study by Sriwong [16]. However, the categorizations of FiND (non-frail, frail, and disabled) are different from that of both FATMPH and FFP (non-frail, pre-frail, and frail), which could affect the sensitivity of the tests and which might be a reason that FiND had the lowest sensitivity in the present study. FATMPH had a higher sensitivity than FiND because it was modified from FFP, but its sensitivity as a screening tool remains poor. In addition, FATMPH and FiND both had high specificity, similar to other tools used in previous studies [15,16]. Most of the screening tools had a specificity of higher than 85%: CSF, at 98.41%, as found in a previous study [15]; and FiND, at 97.39%, as found in the present study. The sensitivity of both FATMPH and FiND were lower than 85%, suggesting that neither is an adequate screening tool [30], while the high specificities of both CSF and FiND suggest they are appropriate for confirming the absence of the condition. FiND is a self-assessment questionnaire suitable for use for individuals in communities, as well as in primary care, whereas FFP is appropriate in primary care and acute care for both individuals in communities and in clinical settings, although the assessment time of FFP is longer than that of FiND [31]. The final judgement of whether or not these methods are appropriate will depend on the context. If the score is used as part of a sequence of screening steps, sensitivity is likely to be more important than specificity, while if the score is used to guide treatment initiation, specificity is equally important [32].

The reliability of FATMPH and FiND were compared with FFP and evaluated using Cohen’s kappa statistic. The kappa values of FATMPH and FiND were 0.289 (95% CI = 0.132–0.445) and 0.147 (95% CI = 0.004–0.241), respectively. The levels of agreement of these values were fair (0.21 ≤ K ≤ 0.40) and slight (0.00 ≤ K ≤ 0.20) [33], respectively. Additionally, in a research context, this measure depends on the prevalence of the condition (with a very low prevalence, κ will be very low, even with high agreement between the raters) [32]. FATMPH’s kappa agreement level was higher than FiND because FATMPH was modified from FFP. Aguayo GA et al. [8], in a study of the agreement between 35 published frailty scores in the general population, found a very wide range of agreement (Cohen’s kappa = 0.10–0.83). The frailty phenotype properties were impacted by the modified frailty phenotype criteria [11]. The prevalence of frailty was 31.2% for modified self-reported walking, 33.6% for modified self-reported strength, and 31.4% for modified self-reported walking and strength [11]. The agreement with the primary phenotype was 0.651 for modified self-reported walking, 0.913 for modified self-reported strength, and 0.441 for modified self-report walking and strength [11]. FATMPH had a lower agreement (0.268) than that of the Modified Frailty Phenotype. We think that the physical inactivity criteria of FATMPH, i.e., the “Can you walk by yourself or do you need someone help you? (no = 0, yes = 1)” should be re-evaluated, as it appears to be very similar to the walk speed criteria (4.5 m walk time; <7 s = 0, ≥7 s or cannot walk = 1). FFP has two measurements (grip strength and walking speed), but FATMPH uses only walking speed and includes fewer detailed questions. Frailty scores show marked heterogeneity because they are based on different concepts of frailty and research results based on different frailty scores cannot be compared or pooled [8].

A limitation of our study is that it was not representative of all community-dwelling older Thais because the participants were all older patients at the OPD of an academic hospital (Maharaj Nakorn Chiang Mai Hospital) and most were urban residents receiving regular government welfare payments. Further study of validated frailty assessment tools such as multicenter studies, as well as other assessment tools, are necessary to ensure their suitability for the Thai population context.

## 5. Conclusions

Our academic hospital-based study using the Thai-language version of the Frailty Assessment Tool of the Thai Ministry of Public Health (FATMPH) and the FiND questionnaire found that both have only a fair to slight agreement with Fried’s Frailty Phenotype (FFP). Additionally, their predictive power is low and, thus, insufficient for frailty detection in a clinical setting. Further multicenter study of these and other assessment tools is needed to improve frailty screening in older Thai populations.

## Figures and Tables

**Table 1 ijerph-20-04020-t001:** Demographic characteristics of participants.

Characteristic	No. (*n* = 251)	Percent (%)
1. Sex		
- Male	98	39.04
- Female	153	60.96
2. Age (years)		
- 60–69	164	65.34
- 70–79	71	28.29
- >80	16	6.37
3. Marital status		
- Married	176	70.12
- Divorced	58	22.71
- Single	14	5.58
- Separated	4	1.59
4. Level of education:		
- Lower than high school	79	31.48
- High school	72	28.69
- Higher than high school	71	28.29
- Other degree	28	11.16
5. Religion		
- Buddhism	243	96.81
- Christianity	8	3.19
6. Occupation		
- Government official	127	50.60
- Business owner	63	25.00
- Farmer	23	9.16
- Employee	19	7.57
- Others	19	7.57
7. Monthly income (baht):		
- >10,000	129	51.39
- 5001–10,000	30	11.95
- 1000–5000	29	11.95
- <1000	63	25.10
8. Source of income:		
- Pension	150	59.76
- Children	55	21.91
- Salary	46	18.33
9. Income sufficiency:		
- Sufficient	170	67.73
- Insufficient	53	21.12
- More than sufficient	28	11.16
10. Payment of medical expenses:		
- Government	223	88.84
- Social welfare	8	3.19
- Individual	20	7.97

**Table 2 ijerph-20-04020-t002:** Health status baseline of participants.

Health Status	No. (*n* = 251)	Percent (%)
1. Body mass index (BMI) (kg/m^2^):		
● <20.00	17	6.77
● 20.00–24.99	98	39.04
● 25.00–29.99	112	44.63
● ≥30.00	24	9.56
2. Underlying medical condition:		
● Yes	232	92.43
● No	19	7.57
3. List of medical conditions:		
● Hypertension	165	65.73
● Dyslipidemia	79	31.46
● Diabetes mellitus	42	16.72
● Hyperuricemia	26	10.35
● Glaucoma/cataract	23	9.16
● Chronic kidney disease	19	7.56
● Benign prostatic hypertrophy	11	4.38
● Coronary artery disease	8	3.18
● Cerebrovascular disease	5	1.98
● Malignancy	2	0.78
● Others	22	8.76
4. Number of medications:		
● 0–1	39	15.53
● 2–3	127	50.59
● >3	85	33.86

**Table 3 ijerph-20-04020-t003:** Comparison of number and percent of frail and non-frail participants using Fried’s Frailty Phenotype, the Frailty Assessment Tool of the Thai Ministry of Public Health, and FiND questionnaire by sex.

Sex	Fried’s Frailty Phenotype		Frailty Assessment Tool of the Thai Ministry of Public Health		FiND Questionnaire	
	Frail	Non-Frail	Frail	Non-Frail	Frail	Non-Frail
Male	2 (9.52%)	96 (41.73%)	15 (34.09%)	83 (40.09%)	2 (20.00%)	96 (39.83%)
Female	19 (90.47%)	134 (58.26%)	29 (65.91%)	12 (59.90%)	8 (80.00%)	145 (60.17%)
Total	21 (100%)	230 (100%)	44 (100%)	207 (100%)	10 (100%)	241 (100%)

**Table 4 ijerph-20-04020-t004:** Comparison of test results of the Frailty Assessment Tool of the Thai Ministry of Public Health, FiND questionnaire, and Fried’s Frailty Phenotype.

Fried’s Frailty Phenotype		Frailty Assessment Tool of the Thai Ministry of Public Health		FiND Questionnaire	
Result	Number (Person)	Positive	Negative	Positive	Negative
Positive	21	12	9	4	17
Negative	230	32	198	6	224
Total	251	44	207	10	241

**Table 5 ijerph-20-04020-t005:** Sensitivity, specificity, positive predictive value (PPV), negative predictive value (NPV), and Cohen’s kappa of the Frailty Assessment Tool of the Thai Ministry of Public Health and FiND questionnaire.

Validity and Reliability	Frailty Assessment Tool of the Thai Ministry of Public Health	FiND Questionnaire
Sensitivity	57.14%	19.05%
Specificity	86.09%	97.39%
PPV	27.27%	40.00%
NPV	95.65%	92.94%
AUC	0.716	0.582
Kappa agreement	0.289	0.147
(95% CI, *p*-value)	(0.132–0.445, <0.001)	(0.009–0.241, <0.001)

## Data Availability

The data are available from the corresponding author upon reasonable request.

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
