# Peer review of "Comparison of Frailty Assessment Tools for Older Thai Individuals at the Out-Patient Clinic of the Family Medicine Department"

_ijerph, 2023, doi:10.3390/ijerph20054020_

Round 1

Reviewer 1 Report

Dear Authors, 

The study is well described. Nevertheless, please add some information in discussion about what is the consequences about the Frailty in elderly people and also, why Frailty Assessment is necessary. Also, according to the results, some politic changes or activities to improve weakness, strength, velocity, ..., are need. So, these measures of Frailty Assessment are necessary to? Try to add some information about the consequence of evaluation.

Reviewer 2 Report

The paper is well written, clear in the objectives and has good flow.

There are minor errors that need to be corrected and in international literature it is better to use the term older person rather than elderly.

There are some areas where the referencing hasn't been included 

You could explain in more depth why you think frieds frailty phenotype is lower in a thai population. Why do you think the FATMPH was only a fair correlation with FFP if it is based on FFP?

My suggested amendments are:

Line 12 – try and use the term older rather than elderly if writing for an international audience

Line 13 doesn’t make sense “conducted results the Frailty Assessment” are you trying to say “ was conducted using the …

Line 14 would flow better if it finished “ and the results were compared with…”

Line 77 state ”of frailty in older” rather than elderly 

Line 78 jumps from talking about prevalence to sensitivity and specificity, this needs clearer explanation or can be confusing to the reader

Try and reword to make clearer 

Line 79 reference – are these all references from Sukriang & Punsawadk – in which case separate with semi colons 

Line 101 change to older

Line 160 (add the number to the reference Boribun

Line 176 pay-ment should be payment

Reviewer 3 Report

the authors tested the viability of some questionnaires in Thailand population by comparing them with a standard one.  There are some concerns that need to be addressed. 

1. The reasons that the authors chose FFP method as the standard one and selected FATMPH and FiND to test should be addressed in background part.

2. The AUC curve is an appropriate method to illustrate the power of testing for different methods.

3. A multicenter investigation would be better to derive a reasonable conclusion, after all, selection bias is too obvious in a single center.

Reviewer 4 Report

Dear Authors and Publishers, I wish you all the best for the year 2023. Thank you for involving me in the review of this article entitled:

 "Comparison of Frailty Assessment Tools for Elderly Thai Individuals at the Out-Patient Clinic of the Family Medicine Department" whose objective was to compare frailty assessment tools for the older adults in a hospital setting. Three tools were used with a sample of 251 aged people. The authors concluded that the Thai version of the Thai Ministry of Public Health Frailty Assessment Tool (FATMPH) and the FiND questionnaire show moderate to slight agreement with the Fried Frailty Phenotype (FFP). However, they have poor predictive power and are insufficient to detect frailty in a clinical setting. They recommend that further study of these and other assessment tools is needed to improve frailty screening in Thai elderly populations. This study has strengths, weaknesses and issues that are highlighted below.

Strengths: The study was validated by ethical clearance. The writing appears to be of a fairly good level of language. The article has the expected classical structure. The literature review seems appropriate.

Weaknesses: The authors point out that one limitation of their study is that it is not representative of Thai community-dwelling aged people, as the participants were all aged OPD patients at a teaching hospital (Maharaj Nakorn Chiang Mai Hospital) and most of them were urban residents receiving regular government welfare benefits. Could this limitation apply to a study conducted in a hospital setting? The results cannot be extended to other hospitals or community-dwelling older people living in the community. They are therefore only applicable to this hospital. The authors should therefore specify in their objective that it is a comparison of frailty screening tools within a hospital setting and mentionned as a limitation that the study is not representative of city hospital settings, which would be more understandable. We would also like to know why the study was not conducted in the community and why the WHO ICOPE tool was not also tested?

The authors should also propose in their conclusion that one or more studies be conducted in hospitals or in the community with representative samples so that the conclusions can be generalised.

In the text, we suggest adding the reference to the definition of frailty (lines 28 and 29) and replacing "Elderly" with "older adult".

Round 2

Reviewer 3 Report

I still insist that multicenter study is necessary to derive a reasonable conclusion.

Author Response

Dear Reviewer 3

From your comment, “ I still insist that multicenter study is necessary to derive a reasonable conclusion.”

We appreciate you sharing your thoughts. Line 337’s final sentence, “Further multicenter study of these and other assessment tools is needed to improve frailty screening in older Thai populations.” has been amended to add the word “multicenter”. If more action is required, kindly let us know.           

Best regards,